# Finding the equilibrium of organic electrochemical transistors

Vikash Kaphle[1], Pushpa Raj Paudel[1], Drona Dahal[1], Raj Kishen Radha Krishnan[1] & Björn Lüssem[1]✉

Organic Electrochemical Transistors are versatile sensors that became essential for the field of organic bioelectronics. However, despite their importance, an incomplete understanding of their working mechanism is currently precluding a targeted design of Organic Electrochemical Transistors and it is still challenging to formulate precise design rules guiding materials development in this field. Here, it is argued that current capacitive device models neglect lateral ion currents in the transistor channel and therefore fail to describe the equilibrium state of Organic Electrochemical Transistors. An improved model is presented, which shows that lateral ion currents lead to an accumulation of ions at the drain contact, which significantly alters the transistor behavior. Overall, these results show that a better understanding of the interface between the organic semiconductor and the drain electrode is needed to reach a full understanding of Organic Electrochemical Transistors.

[1] Department of Physics, Kent State University, Kent, OH 44242, USA. ✉email: blussem@kent.edu

The organic electrochemical transistor (OECT) is a key device for many bio-electronic applications as it is able not only to transduce ionic into electronic signals but also to amplify chemical signals close to their source, ensuring a high signal-to-noise ratio. Conducting polymers such as PEDOT:PSS are almost exclusively used as active material of OECTs, which not only show a balanced charge carrier and ion mobility, but are often bio-compatible and flexible as well[1–3].

In PEDOT:PSS-based transistors[4], cations are injected from the electrolyte into the PEDOT:PSS layer, which neutralizes the PSS$^-$ groups and de-dopes the PEDOT:PSS layer. The amount of cations injected is controlled by the potential applied across the electrolyte acting as a gate. Hence, the conductivity of the PEDOT:PSS film measured between two electrodes (source and drain) is controlled by the potential on a third, the gate electrode[5].

Several models were proposed to quantitatively describe OECT behavior, in particular the process of doping/de-doping the organic semiconductor[6–13]. The most widely used models separate the device into an ionic and electronic system[3,14,15]. Ions are assumed to move vertically (i.e., from the gate electrode into the transistor channel, in the following denoted as $y$-axis), whereas hole transport is restricted to a horizontal movement from source to drain electrode (i.e., along the $x$-axis). This assumption, resembling the gradual channel approximation of standard thin-film theory, allows to calculate the density of ions inside the transistor channel $p_{ion}(x)$ as a function of the difference between the channel potential $\Phi(x)$ and the applied gate potential $V_{GS}$[3,8,15].

To be able to derive an analytic description of OECTs, it is often postulated that the density of injected ions $p_{ion}(x)$ is directly proportional to the potential difference $V_{GS} - \Phi(x)$. Under this assumption, the process of injecting ions into the channel can be described by a capacitive element $C_G$ included between the gate electrode and the PEDOT:PSS channel. In a first version of the model, the capacitance was assumed to scale with device area[8], whereas Rivnay et al.[16] found that it depends on the volume of the semiconductor channel. This volumetric gate capacitance reflects the observation that ions are injected into the full volume of the polymer, resulting in huge transconductance values observed frequently[17].

Capacitive models are widely used in the literature[3,14,15] as they allow to conveniently discuss and analyze transistor results. However, the precise nature of the gate capacitance is still intensively discussed. Some model relax the assumption of a direct proportionality between $V_{GS} - \Phi(x)$ and $p_{ion}(x)$ and instead calculate the density of injected ions from basic drift–diffusion equations. For example, Shirinskaya et al.[18] used a 1D numerical model to determine $p_{ion}(x)$ and hence the conductivity of the organic semiconductor as a function of position inside the transistor channel $\sigma(x)$. Coppedè et al.[19] proposed an analytical 1D solution for the ionic current injected into PEDOT:PSS under the assumption of a constant electric field inside the electrolyte.

However, regardless of the detail with which the density of injected ions $p_{ion}(x)$ is calculated, ion movement was always limited to one dimension, that is, to a movement perpendicular to the transistor channel. This assumption, however, has been put into question by recent results of Szymanski, who found that not restricting ion movement inside the transistor to one dimension leads to a different ion concentration $p_{ion}(x)$ as predicted by capacitive models[11].

Here, we study the applicability of capacitive models by analyzing the electric potential along the transistor channel $\Phi(x)$. Our data indicate that these models indeed fail to describe the steady state of the transistors. It is shown that the assumption of a gate capacitance leads to ion concentrations inside the transistor channel that would result in significant lateral ion currents. These lateral currents, however, are neglected in capacitive models, forcing the derived solutions into an unrealistic, out-of-equilibrium state. With the help of a 2D drift–diffusion model that solves the continuity equation of holes and cations consistently along the $x$- and $y$-direction, it is shown that in contrast to predictions of current OECT models, ions follow an exponential distribution along the transistor channel, which leads to an accumulation of ions at the drain electrode and an additional potential drop at the interface. Overall, the newly found steady-state distribution of ions inside the transistor channel shifts the focus to understand details of device operation away from the bulk organic semiconductor to the organic semiconductor–drain electrode interface.

## Results

### Calculating the potential along the channel from capacitive models.
In contrast to the output and transfer characteristics, the potential distribution along the transistor channel provides more detailed, spatially resolved information that can be used to test the predictions of device models.

Capacitive device models as sketched in Fig. 1 allow to analytically describe the transistor behavior and in particular to calculate the potential along the channel $\Phi(x)$. These models, originally proposed by Bernards and Malliaras[8] and Rivnay et al.[16] consist of a distributed ionic resistance $R_{ion}$, describing ion transport inside the electrolyte, a gate capacitance $C_G$, and the PEDOT:PSS layer.

The gate capacitance $C_G$ is used to determine the amount of cations $p_{ion}(x)$ inside the PEDOT:PSS layer. It is assumed that $p_{ion}(x)$ is directly proportional to the potential difference $\Delta V(x) = V_{GS} - \Phi(x)$, where $V_{GS}$ is the potential applied to the gate. This step implicitly invokes the gradual channel approximation

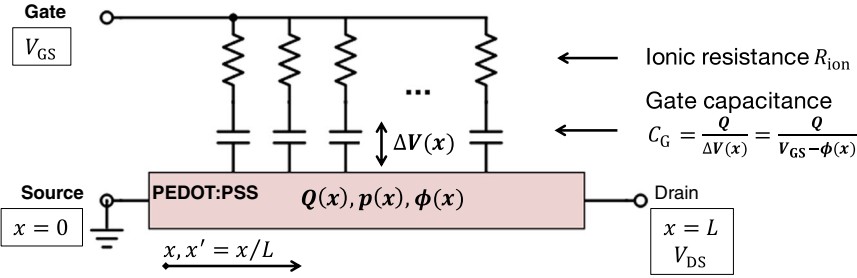

**Fig. 1 Equivalent circuit used in capacitive device models.** As firstly proposed by Bernards and Malliaras[8], a gate capacitance $C_G$ is used to calculate the ion concentration $Q$ inside the transistor channel. Based on this assumption, an analytical solution for the hole concentration $p(x)$, the channel potential $\Phi(x)$, and the drain current $I_D$ as a function of the gate and drain potential $V_{GS}$ and $V_{DS}$ can be found.

used in thin-film transistor theory (cf. Supplementary Note 3). It follows:

$$p_{ion}(x) = \frac{C_G}{e}(V_{GS} - \Phi(x)),\qquad(1)$$

where $C_G$ is given in $Fcm^{-3}$[16]. The injected cations de-dope the PEDOT:PSS layer. Assuming charge neutrality, the density of free holes $p(x)$ becomes

$$p(x) = p_0 - p_{ion}(x) = p_0 - \frac{C_G}{e}(V_{GS} - \Phi(x)),\qquad(2)$$

where $p_0$ is the density of holes in the PEDOT:PSS layer without injected cations, that is, the doping concentration, which is supposed to be proportional to the density of sulfonate (PSS$^-$) groups.

The charge carrier mobility in PEDOT:PSS $\mu$ was observed to depend on the hole concentration $p(x)$[20]. Such a dependency is well known from other polymer or small-molecule semiconductors[21] and is explained by a large energetic disorder in the polymer. Friedlein et al.[20] proposed the following relation to describe the hole dependency of the carrier mobility $\mu(p)$ ($k_B$: Boltzmann's constant, $T$: temperature, $\mu_0$: hole mobility at $p = p_0$):

$$\mu(p) = \mu_0\left(\frac{p}{p_0}\right)^{\frac{E_0}{k_BT}-1},\qquad(3)$$

In Eq. (3), the strength of disorder is described by the energy $E_0$, which describes by how far energetic states tail into the gap of the material. A good fit to the experimental data was obtained for $\frac{E_0}{k_BT} \approx 2$.

As shown by Friedlein et al.[20] the drain current $I_D$ in the linear regime of the transistor at a particular gate and drain potential $V_{GS}$ and $V_{DS}$ can be derived by integrating Ohm's law from the source (at $x = 0$) to the drain electrode (at $x = L$). With the help of Eq. (3), one obtains

$$\int_0^L j\,dx = \int_0^{V_{DS}} -ep(x)\mu(x)\,d\Phi,\qquad(4)$$

$$I_D = p_0 e\mu_0\frac{wd}{L}\frac{V_P}{\frac{E_0}{k_BT}+1}\left\{\left[1 - \frac{V_{GS}-V_{DS}}{V_P}\right]^{\frac{E_0}{k_BT}+1} - \left[1 - \frac{V_{GS}}{V_P}\right]^{\frac{E_0}{k_BT}+1}\right\}.\qquad(5)$$

Here, $e$ is the elementary charge, $w$ is the width of the transistor, $d$ is the thickness of the PEDOT:PSS layer, and $V_P$ is the pinch-off voltage of the transistor. For a constant mobility $\left(\frac{E_0}{k_BT} = 1\right)$, one recovers the original result found by Bernards and Malliaras[8] and Rivnay et al.[16],

$$I_D = \frac{wd}{L}\mu_0 C_G\left[V_P - V_{GS} + \frac{1}{2}V_{DS}\right]V_{DS}.\qquad(6)$$

The potential along the channel $\Phi(x)$ in the linear region of the transistor can be obtained by integrating Ohm's law not all the way from the source to the drain electrode, but from the source (at $x = 0$) to any position $x$ inside the channel, that is,

$$\int_0^x j\,dx = \int_0^{\Phi(x)} -ep(x)\mu(x)\,d\Phi.\qquad(7)$$

Furthermore, with the help of Eq. (5), one obtains for a constant mobility ($\frac{E_0}{k_BT} = 1$, see Supplementary Note 1)

$$\Phi(x') = (V_{GS} - V_P)$$
$$+ \sqrt{[V_{DS} - (V_{GS} - V_P)]^2 x' - (V_P - V_{GS})^2(x' - 1)},\qquad(8)$$

and for $\frac{E_0}{k_BT} = 2$,

$$\Phi(x') = (V_{GS} - V_P)$$
$$+ \sqrt[3]{\left\{[V_{DS} - (V_{GS} - V_P)]^3 x' - (V_P - V_{GS})^3(x' - 1)\right\}}\qquad(9)$$

For both cases, $x' = \frac{x}{L}$ is the $x$-coordinate scaled by the total length of the channel $L$.

As seen in Eqs. (8) and (9), apart of the externally applied voltages $V_{GS}$ and $V_{DS}$, the potential inside the channel is controlled by the pinch-off voltage $V_P$ only. The pinch-off voltage $V_P$ describes the depletion of the doped channel. The potential difference applied across the gate capacitance $\Delta V(x)$ is largest close to the drain and hence most ions $p_{ion}(x)$ accumulate at the drain electrode. Hence, depletion of the channel is strongest at the drain (i.e., $p(x)$ is smallest, Eq. (2)). The pinch-off voltage can be understood as the potential difference $\Delta V(x = L) = V_{GS} - V_{DS}$ at which the PEDOT:PSS layer is fully depleted at the drain, that is, $p(x = L) = 0$. It follows from Eq. (2):

$$\Delta V(x = L)|_{\text{pinch off}} = V_{GS} - V_{DS} = V_P = \frac{ep_0}{C_G}.\qquad(10)$$

Once the channel is fully depleted at the drain, the drain current saturates. Hence, the saturation voltage $V_{DS,sat}$ becomes $V_{DS,sat} = V_{GS} - V_P$.

**Measuring the potential along the transistor channel.** An OECT design with five potential probes as shown in Supplementary Fig. 1 is used to measure the potential along the transistor channel. Gold electrodes are deposited by thermal evaporation and structured by photolithography. PEDOT:PSS is spin coated onto the electrodes and etched by oxygen plasma using a shadow mask to remove the semiconductor outside the gate and channel area. The electrolyte is prepared by mixing the room temperature ionic liquid (1-ethyl-3-methylimidazolium ethyl sulfate (C2MIM EtSo4)) and 100 mM sodium chloride in a 4:1 ratio. The electrolyte is placed on top of the gate and channel area. All devices are characterized inside a glovebox (Nitrogen). Process details are given in the "Methods" section. Four identical samples are processed for every data point. The results shown in the following are obtained for devices with a PEDOT:PSS layer thickness of 180 nm, unless otherwise noted.

The design of the transistors (cf. Supplementary Fig. 1) allows to measure the potential at five positions within the channel. Drain and source electrodes are biased at a particular voltage at the two outer contacts and the gate potential is applied at the gate contact. In the following, only the section of PEDOT:PSS not covered by source–drain electrodes is counted toward the channel length, that is, sections underneath the electrodes do not add to the total channel length. However, as the potential in Eqs. (8) and (9) only depends on the scaled position inside the transistor channel $x' = \frac{x}{L}$, counting the sections of the channel underneath the electrodes as well will not change conclusions drawn later. The applied gate voltage ranges from $-0.1$ to 0.7 V and the drain voltage is varied from $-0.1$ to $-0.5$ V, so that the linear as well as the saturation region of the transistor is studied. Figure 2 plots the measured average channel potential vs. position for varying drain voltages $V_{DS}$ at $V_{GS} = -0.1$ V (Fig. 2a) and $V_{GS} = 0.7$ V (Fig. 2a). The remaining plots for $V_{GS} = 0.1, 0.3,$ and 0.5 V are shown in Supplementary Fig. 3. At $V_{GS} = -0.1$ V, the transistor is operated in the linear regime. Consequently, the potential profile increases steadily towards the drain contact. However, the potential profile is not perfectly linear, but there seems to be an additional voltage drop between the source (at $x = 0$) and the first

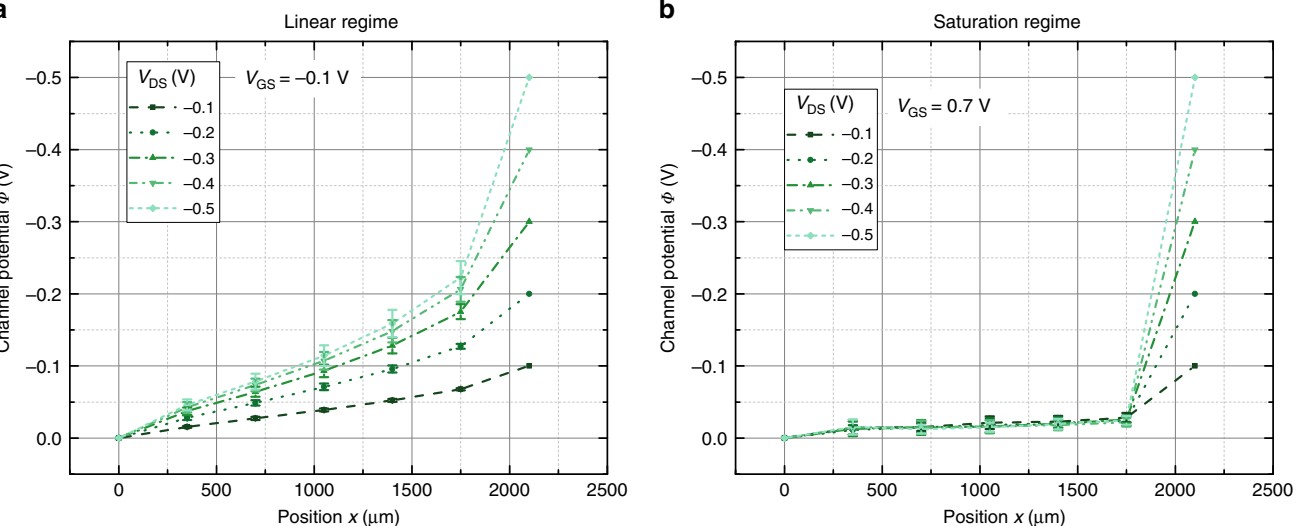

**Fig. 2 Measured channel potential. a** Potential along the transistor channel in the linear regime (at $V_{GS} = -0.1$ V). **b** Potential in the saturation regime (at $V_{GS} = 0.7$ V). The error bars display the standard deviation of four identical samples.

potential probe, and between the last potential probe and the drain (at $x = L$).

In contrast, the transistor is in saturation at $V_{GS} = 0.7$ V. The potential remains almost constant inside the channel and most potential drops at the drain contact, which reflects the pinch off of the channel close to the drain.

Figure 3a shows the result of the fit of the channel potential by Eq. (8), that is, under the assumption that the charge carrier mobility of PEDOT:PSS is independent of the charge carrier concentration. Only the results for $V_{GS} = -0.1$ V are shown in Fig. 3a, the remaining fits are given in Supplementary Fig. 4. Care has been taken to restrict the fit to the linear regime of the transistors, that is, not to extend the fit beyond the validity of Eq. (8).

Equation (8) has only one free parameter, the pinch-off voltage $V_P$, which according to capacitive models discussed above is entirely determined by design parameters, that is, $V_P = \frac{qp_0}{C_G}$. Therefore, the pinch-off voltage is expected to be independent of the applied drain and gate potential. However, fitting all potential profiles in the linear region by Eq. (8) is not feasible if the pinch-off voltage is kept constant. Instead, to obtain reasonable fits, the pinch-off voltage has to be adapted for every gate and drain potential. Still, the fits using Eq. (8) display a systematic underestimation at the fifth potential probe at $x' = 0.83$, which increases for increasing drain potential.

The extracted pinch-off voltages are plotted with respect to the drain and gate bias in Fig. 3c. We restrict our fit to the linear region, that is, to the range $V_{GS} - V_{DS} < V_P$ to ensure that the potential is not affected by the pinch off at the drain contact. The straight line in the figure separates the linear region and saturation region of the transistor. Overall, the extracted pinch-off voltages $V_P$ increase with increasing drain potential and gate potential.

The quality of the fit is slightly increased when a non-constant charge carrier mobility as defined by Eq. (3) (i.e., for $\frac{E_0}{k_B T} = 2$) is used. In Fig. 3b, the result of the fit using Eq. (9) is shown for $V_{GS} = -0.1$ V (cf. Supplementary Fig. 5 for other voltages). Although the fit is slightly improved, the potential at $x' = 0.83$ is still underestimated. Similar to the results obtained for a constant mobility (Fig. 3a), a satisfying fit can only be obtained when adjusting the pinch-off voltage for the different gate and drain potentials. The average pinch-off voltage $V_P$ obtained from the fit

of four devices is plotted in Fig. 3c. Overall, the pinch-off voltage strongly depends on the gate potential and the drain potential. Whereas the dependency of $V_P$ on the drain potential is weak for larger negative voltages, it is pronounced for $V_{DS} > -0.3$ V.

**Capacitive OECT models do not describe the equilibrium state of OECTs.** The fits shown in Figs. 3a, b could only be obtained by using the pinch-off voltage $V_P$ as a fit parameter and adjusting $V_P$ for every gate and drain potential. However, considering that $V_P = \frac{qp_0}{C_G}$ (Eq. (10)), the pinch-off voltage is defined by design parameters only and should not vary with the applied potential. This failure of the model to consistently describe the potential along the transistor channel indicates that the model does not describe device operation correctly. Indeed, in the following it is shown that it does not describe the steady state of the device.

In capacitive device models, the density of cations is calculated by the gate capacitance, that is, using Eq. (1). This approach implicitly assumes that cations entering the channel from the electrolyte do not move laterally, which, however, is in contradiction with the moving front experiments by Stavrinidou et al.[22] showing that cations move efficiently inside the PEDOT:PSS. Considering the considerable ion mobility inside PEDOT:PSS, cations do not only move vertically from the gate electrolyte into the channel, but will as well move laterally from source to drain under the influence of the lateral source–drain field.

Neglecting lateral ion movement forces the device into a non-steady-state configuration, which is displayed in Fig. 4. Here, the normalized hole density $p(x)/p_0$, the normalized density of cations $p_{ion}/p_0$ and the electric field inside the channel as obtained for a hypothetical OECT of channel width $W = 1$ mm, channel thickness $d = 200$ nm, channel length $L = 100$ μm, doping concentration inside the PEDOT:PSS layer of $p_0 = 10^{20}$ cm$^{-3}$, mobility $\mu_{hole} = 0.25$ cm$^2$ (Vs)$^{-1}$, gate voltage, $V_{GS} = 0.2$ V, and drain voltage $V_{DS} = -0.5$ V are plotted. Details of the equations used for Fig. 4 are given in Supplementary Note 7. All calculations are based on Eq. (1), that is, on the assumption of a gate capacitance.

In Fig. 4a, one observes that the hole density $p(x)$ (calculated from Supplementary Eq. 5) decreases from the source at $x = 0$ to the drain at $x = L = 100$ μm. At the same time, the electric field $E$ increases from source to drain. Overall, the hole drift current $j_h = e\mu p(x)E(x)$ plotted in Fig. 4b remains constant, which reflects

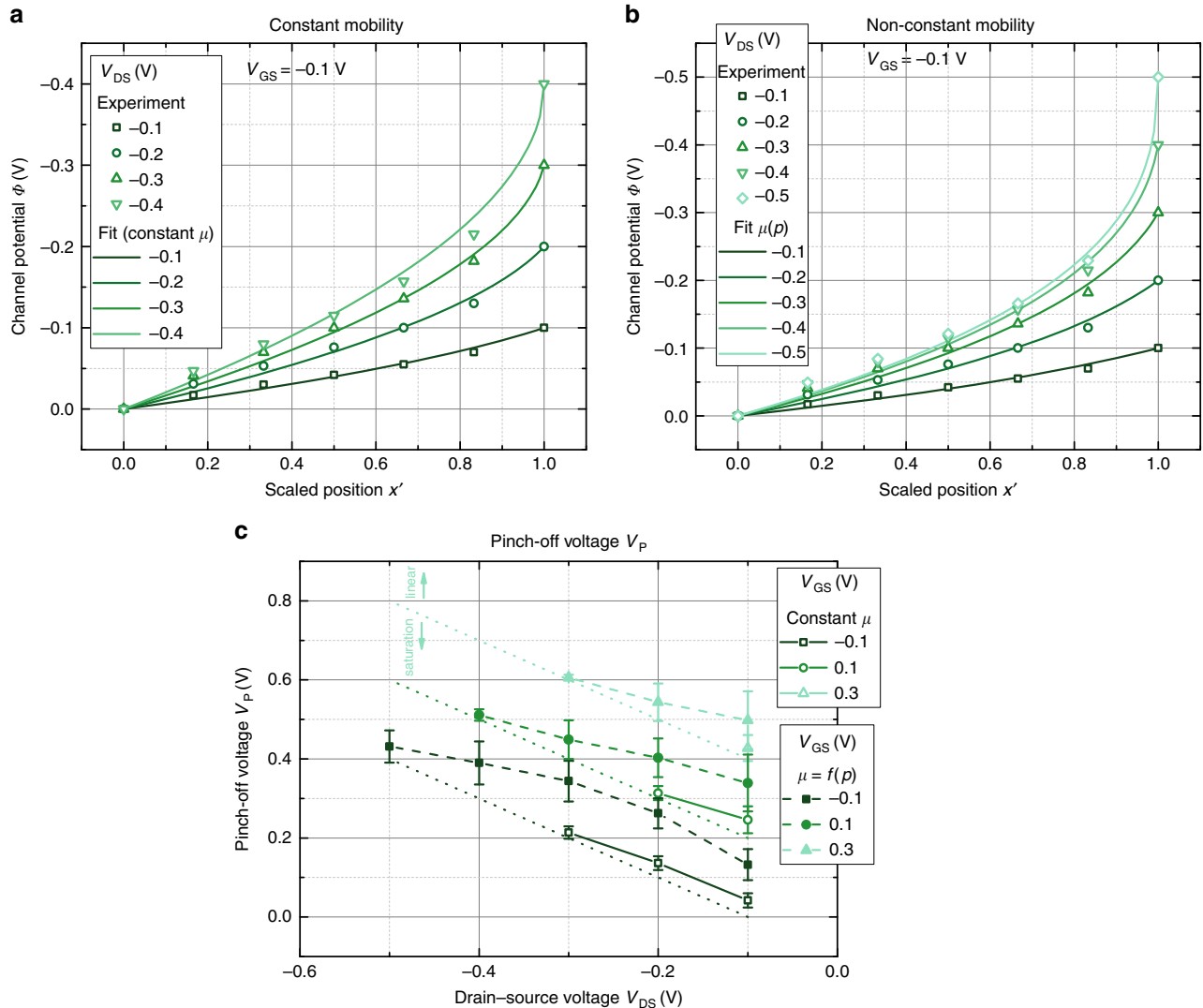

**Fig. 3 Fitting the channel potential. a** Channel potential $\Phi(x')$ at $V_{GS} = -0.1$ V fitted by Eq. (8), that, assuming a constant mobility. **b** Channel potential $\Phi(x')$ at $V_{GS} = -0.1$ V fitted by Eq. (9), that is, assuming a mobility that depends on the hole concentration (Eq. (3) with $\frac{E_0}{k_B T} = 2$). **c** Pinch-off voltage extracted from the fits for constant mobility (open symbols) and a mobility that depends on the hole concentration (closed symbols). The dotted lines denote the transition form of the linear to the saturation regime of the transistor for the different gate voltages. A satisfying fit can only be found if the pinch-off voltage is adjusted for every applied potential.

the fact that all current injected at the source has to leave the device at the drain electrode.

The density of ions inside the channel $p_{ion}$, calculated from Eq. (2) is plotted in Fig. 4a as well. Similar to the hole concentration, the ion concentration $p_{ion}$ increases from the source at $x = 0$ to the drain electrode at $x = L$. This increase in ion concentration is caused by the increasing difference between the gate and the channel potential, that is, a larger voltage is applied across the gate capacitance.

The ion current is given by the sum of drift current $j_{ion,drift} = e p_{ion}(x) \mu_{ion}(x) E(x)$ and diffusion current $j_{ion,diff} = -e D_{ion} \frac{d p_{ion}(x)}{dx}$, both of which are plotted in Fig. 4b. Here, it is assumed that the diffusion constant $D_{ion}$ is related to the ion mobility $\mu_{ion}$ by Einstein's equation. Throughout the channel, neither the total current nor the drift or diffusion currents are constant. This shows that current OECT models represent a non-equilibrium state of the device. Or, in other words, the model yields an unrealistic ion distribution inside the transistor channel, which would result in the generation of arbitrary ion currents inside the channel. This

problem is inherent to the model; it is caused by assumption that the ion concentration can be determined by a gate capacitance, that is, Eq. (1), which forces the ion distribution into a state far from equilibrium.

The calculation of the ion current discussed above implicitly assumes that the PEDOT:PSS layer is homogeneous, that is, ions and holes experience an identical electric field $E(x)$. However, Volkov et al.[13] recently argued that the PEDOT:PSS layer has to be treated as a mixture of an ion-conductive phase formed by the PSS⁻ groups and a hole conductive phases consisting of PEDOT crystallites. Caused by this phase separation, two distinct electrical potentials are established in the active layer—one potential inside the PSS⁻ and one inside the PEDOT phase[12,13].

Still, despite the assumption of two electric potentials inside the semiconductor layers, the general argument given here remains valid, although a closed analytical solution cannot by derived. The PSS⁻ phase is in contact with the source and drain electrode, that is, it is subject to a lateral electric field. Restricting the flow of ions to the vertical direction, that is, by setting the potential of

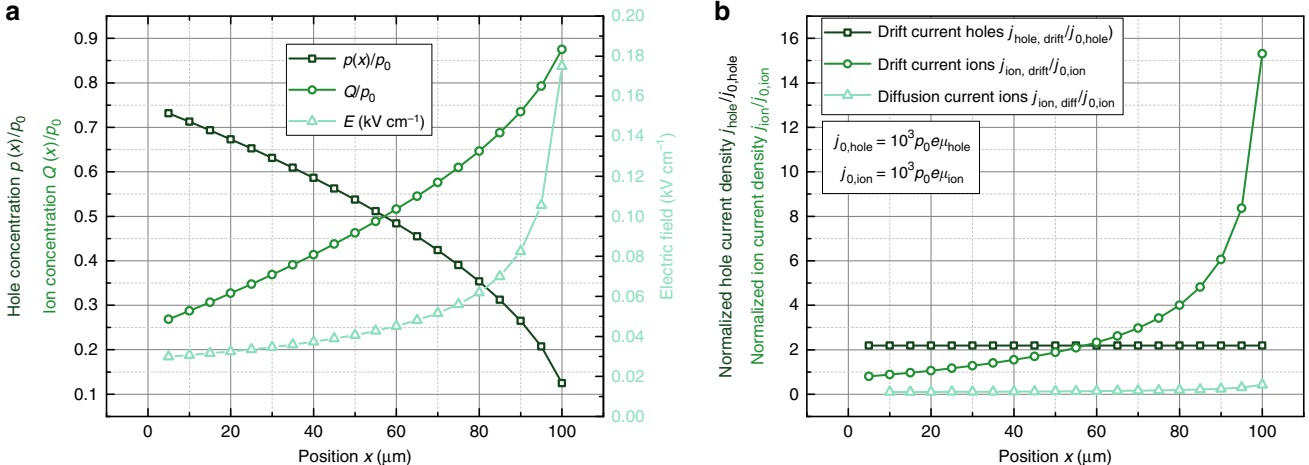

**Fig. 4 Lateral ion currents. a** Normalized hole density, ion density, and electric field inside a hypothetical p-type OECT ($W = 1$ mm, $d = 200$ nm, $L = 100$ μm, $p_0 = 10^{20}$ cm$^{-3}$, $\mu_{hole} = 0.25$ cm$^2$ (Vs)$^{-1}$, $V_{GS} = 0.2$ V, $V_{DS} = -0.5$ V). **b** Although the normalized hole current is constant across the channel, the ionic drift and diffusion current increases, which indicates that the current model leads to a non-steady-state solution.

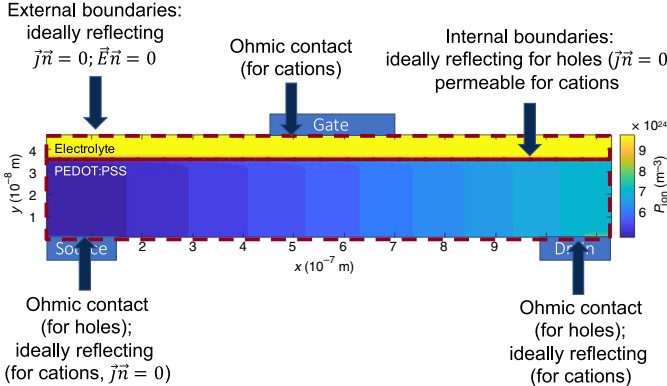

**Fig. 5 Setup of 2D OECT model.** Drift–diffusion model used to clarify the steady-state behavior of OECTs. The OECT is modeled by a layer of PEDOT:PSS, covered by an electrolyte. The PEDOT:PSS layer is contacted by source and drain electrodes on the bottom, which are treated as Ohmic contacts for holes, but are ideally reflecting for ions, that is, the normal ion current is set to zero at the electrodes. The electrolyte is contacted by the gate electrode, which is treated as ohmic for ions, that is, the cation concentration is set to its equilibrium concentration at the interface. Holes can only migrate inside the PEDOT:PSS layer, but cations can migrate inside the electrolyte and PEDOT:PSS. Anions inside the electrolyte and PSS$^-$ ions in the PEDOT layer are assumed to be stationary.

the PSS$^-$ to a constant value[12], leads to a non-steady-state distribution of ions.

**2D modeling to obtain the equilibrium state of OECTs.** In order to obtain the steady-state ion distribution inside the transistor channel, the continuity equation of holes and ions have to be solved consistently in *both* dimensions, $x$ and $y$. Treating both continuity equations on an equal footing will allow lateral ion currents to equilibrate, that is, the system will reach an equilibrium configuration.

In Fig. 5, the setup of the 2D drift–diffusion simulation used to discuss OECT operation is shown. The device consists of a source and drain electrode located on the bottom of the device, followed by a layer of PEDOT:PSS on top. The PEDOT:PSS layer is covered by a layer of electrolyte, which is contacted by a gate electrode on top. The simulation routine was implemented based on a finite difference discretization scheme[23]. Poisson's equation

$$\nabla^2 \Phi(x,y) = \frac{e}{\epsilon \epsilon_0} \left( -p(x,y) - p_{ion}(x,y) + p_0(y) + N_0(y) \right), \quad (11)$$

and the continuity equation for holes and cations in steady state

(i.e., setting all time derivatives to zero)

$$\nabla \overrightarrow{\mathbf{j_p}} = 0, \quad (12)$$

$$\nabla \overrightarrow{\mathbf{j_{p,ion}}} = 0 \quad (13)$$

are solved self-consistently in a Gummel scheme. Here, $N_0$ is the concentration of anions in the electrolyte, and $\epsilon$ is the dielectric constant in the device. Using Eq. (11), we implicitly assume that the dielectric constant is constant throughout the device. Furthermore, we neglect any effects due to migration of anions inside the electrolyte or the PEDOT:PSS layer. It is assumed that the problem is satisfactorily approximated by Boltzman statistics, that is, the Einstein equation $D = \mu V_T$ is used for holes and cations ($D$ and $\mu$ are the diffusion constant and the mobility of either holes or cations, and $V_T = \frac{k_B T}{e}$ is the thermal voltage). Furthermore, the voltage applied externally to the gate electrode $V_{GS} = V_{GS,int} + \Delta\mu$ is offset by the difference in the chemical potential of the ohmic gate contact and the source contact $\Delta\mu$, leading to an internal voltage $V_{GS,int}$. More details about the algorithm are given in Supplementary Note 4.

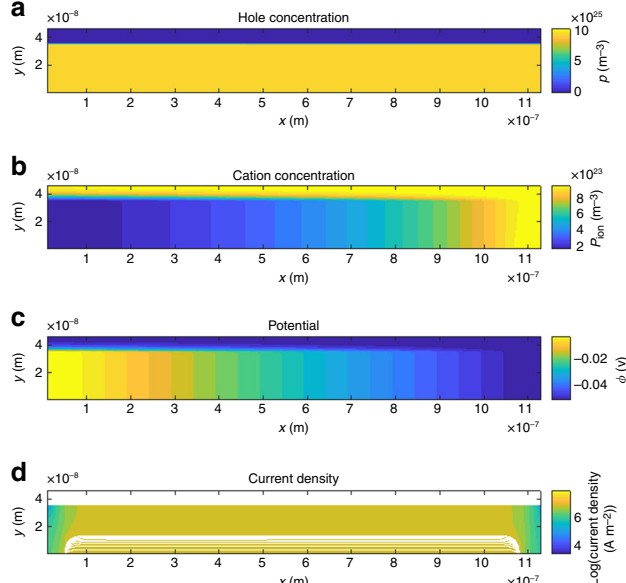

**Fig. 6 Simulation results for short-channel OECTs. a** Hole concentration $p(x, y)$, **b** cation concentration $p_{ion}(x, y)$, **c** electric potential $\Phi(x, y)$, and **d** hole current density $j_p(x, y)$ calculated using the parameters summarized in Supplementary Table 1 at $V_{DS} = -2V_T$ and $V_{GS,int} = -2V_T$. Streamlines are shown in the bottom panel indicating the current flow from the source electrode on the left to the drain electrode on the right.

Whereas ions can migrate in the whole device volume, holes are restricted to the PEDOT:PSS layer. The source and drain electrodes are modeled as ohmic contacts for holes, that is, the boundary condition for holes at the boundaries is set to the doping concentration inside the PEDOT:PSS layer $p_0$. For cations, source and drain electrodes are treated as ideally reflecting, that is, the cation current normal to the electrode $\overrightarrow{\mathbf{j}_{ion}} \cdot \overrightarrow{\mathbf{n}}$, with $\overrightarrow{\mathbf{n}}$ the surface normal of the interface, is set to zero. The interface between PEDOT:PSS and electrolyte is treated as ideally reflecting for holes (forcing the vertical hole currents at the top of the PEDOT:PSS layer to zero), and permeable to cations. The gate electrode is modeled as an ohmic contact for cations, that is, the cation concentration is set to the anion concentration $N_0$.

Simulation results are shown in Fig. 6. The parameters used for the device are summarized in Supplementary Table 1. The device has a channel length $L = 1.13$ μm, the PEDOT:PSS layer is 31 nm thick, an ion concentration of $N_0 = 10^{18}$ cm$^{-3}$ and a doping concentration inside the PEDOT:PSS layer of $p_0 = 10^{20}$ cm$^{-3}$ is assumed. Figure 6 plots the hole concentration, cation concentration, electric potential, and hole current at a drain potential $V_{DS} = -2V_T$ and a gate potential of $V_{GS} = -2V_T$, with $V_T$ the thermal voltage.

The cation concentration plotted in Fig. 6 shows that ions are migrating into the PEDOT:PSS layer. As the electric potential is most negative at the drain electrode at the bottom right of the device, the density of cations is highest at the drain electrode. However, the density of cations is still smaller than the hole density inside the PEDOT:PSS layer, that is, de-doping by cations is weak. Therefore, the hole concentration inside the PEDOT:PSS layer remains at its equilibrium concentration of $10^{20}$ cm$^{-3}$ and the PEDOT layer is highly conductive. Consequently, the electric potential drops almost linearly between the source electrode on the bottom left and the drain electrode at this particular choice of applied potentials.

The cation concentration $p_{ion}(x, y = 0)$, hole concentration $p(x, y = 0)$, and electric potential $\Phi(x, y = 0)$ at a gate potential of

$V_{GS} = -2V_T$ at the bottom of the device (i.e., at $y = 0$) is plotted for varying drain potentials $V_{DS} = 0.. - 8V_T$ in Fig. 7.

It can be seen that the cation concentration increases exponentially toward the drain electrode. This steep increase in cation concentration along the transistor channel is caused by the assumption of vanishing ion currents $j_{p,ion}$, that is, it can be derived analytically assuming that the ion concentration has reached its equilibrium distribution. Assuming that the x-component of the ion current $j_{x,ion} = -ep_{ion}\mu_{p,ion}\frac{d\Phi(x)}{dx} - eD_{p,ion}\frac{dp_{ion}}{dx}$ equals zero and using the Einstein equation leads to

$$-\frac{d\Phi(x)}{dx} = \frac{V_T}{p_{ion}}\frac{dp_{ion}}{dx}. \qquad (14)$$

This differential equation is solved by

$$p_{ion} = N_0 \exp\left(-\frac{\Phi(x) - V_{GS}}{V_T}\right), \qquad (15)$$

that is, assuming that $\Phi(x)$ varies slowly along the channel, $p_{ion}(x)$ is indeed expected to increase exponentially toward the drain electrode, in accordance to the trends observed in Fig. 7a.

The cations inside the channel de-dope the PEDOT:PSS layer and, assuming that the layer remains electrically neutral, the hole concentration $p(x)$ becomes

$$p(x) = p_0 - p_{ion} = p_0\left(1 - \frac{N_0}{p_0}\exp\left(-\frac{\Phi(x) - V_{GS}}{V_T}\right)\right). \qquad (16)$$

Figure 7b plots the hole concentration as obtained by the simulation along the channel. Indeed, $p(x)$ follows the general trend of Eq. (16), that is, the hole concentration drops proportional to $p(x) \propto 1 - \exp\left(\frac{x}{x_0}\right)$. At a gate voltage of $V_{GS} = 8V_T$, the cation concentration almost equals the doping concentration at the drain, the hole concentration is close to zero at the drain, and the channel is pinched off.

The channel pinch-off leads to a saturation in the drain current, seen in the output characteristic of the simulated OECT shown in Supplementary Fig. 2. Channel pinch off will occur at $p_{ion}(x = L, y = 0) = p_0$. Therefore, the transistor will saturate earlier for larger gate voltages, that is, for a higher concentration of cations at the drain (cf. Eq. (15)).

Finally, Fig. 7c shows the potential profile along the channel $\Phi(x, y = 0)$ obtained by the simulation. In contrast to the result derived for the standard OECT model (cf. Eqs. (8) and (9)), the potential increases linearly along the channel. However, at the drain electrode, the potential drops rapidly, which can be explained by the accumulation of cations at the drain seen in Fig. 7a. Most interestingly, this potential step at the drain is not only visible in the saturation regime, but is present for the device in the linear regime as well, although to a smaller extent.

The linear increase in potential along the channel is a direct consequence of the equilibrium ion concentration described by Eq. (15). Starting from the general description of the hole current as a sum of drift and diffusion currents $j_{p,x} = -e\left(p(x)\mu_p\frac{d\Phi(x)}{dx} + D_p\frac{dp}{dx}\right)$, and using Eq. (16), one arrives at

$$j_{p,x} = -e\mu_p N_A \frac{d\Phi(x)}{dx}. \qquad (17)$$

As the hole current along the channel $j_{p,x}$ is constant at steady-state conditions, the gradient in the electric potential is constant as well, that is, the potential rises linearly within the transistor channel.

**Comparison of 2D modeling to experimental results.** The numerical model can be used to fit experimental data. A fit of the transfer characteristic of OECTs with different channel lengths

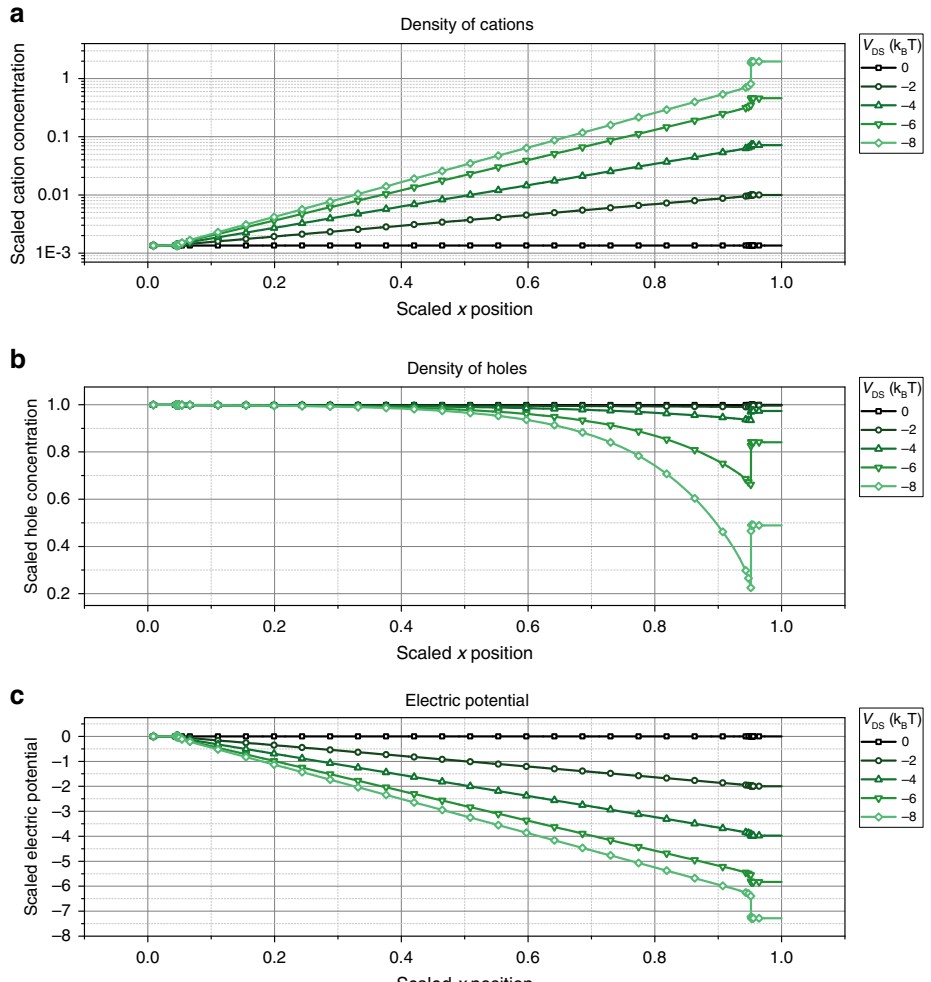

**Fig. 7 Equilibrium state of the transistor channel. a** Density of cations $p_{ion}(x, y = 0)$, **b** holes $p(x, y = 0)$, and **c** electric potential $\Phi(x, y = 0)$ along the channel of the simulated OECT shown in Fig. 6 at $V_{GS} = -2V_T$. The parameters used for the simulation are summarized in Supplementary Table 1. Concentrations are scaled to the doping concentration in the PEDOT:PSS layer $p_0 = 10^{20}$ cm$^{-3}$, spatial coordinates are scaled by the channel length of the device $L_0 = 1.13$ μm, and the potential is scaled by the thermal voltage $V_T = k_B T/e$.

($L = 350, 700, 1050, 1400, 1750,$ and $2100$ μm) is shown in Fig. 8. The parameters used to fit the data are summarized in Supplementary Table 2. Overall, a good agreement between the model and the experimental data is found.

It was shown in Fig. 4 that models based on the assumption of a gate capacitance (e.g., Eq. (2)) cannot explain the potential distribution inside the channel consistently. The numerical model presented here is able to alleviate this contradiction and provides an alternative fit for the channel potential. As seen in Fig. 7c and discussed in Eq. (17), the numerical model predicts a linear increase of the channel potential and an additional potential drop at the channel/drain electrode caused by the accumulation of cations in at the drain.

Figure 9 plots the channel potential as obtained by the 2D drift–diffusion model and compares the results to the measured potential. The same parameters as for Fig. 8 (summarized in Supplementary Table 2) are used. Indeed, the model can explain the experimental trends qualitatively. The potential drop inside the channel is approximately linear within the channel, but jumps at the drain electrode due to the accumulation of cations and the corresponding space charge layer. This result is as well in good

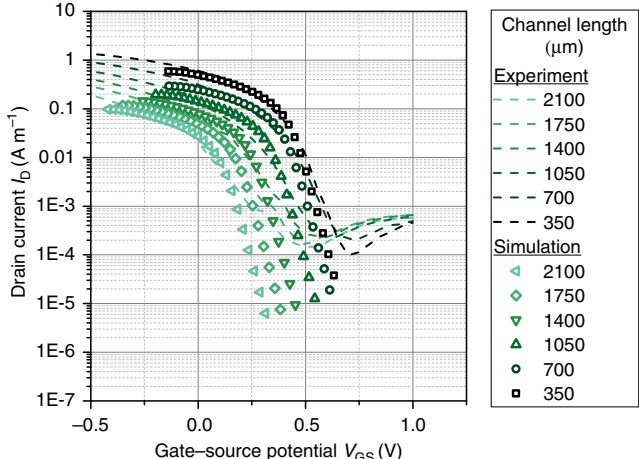

**Fig. 8 Transfer characteristic of OECTs.** Comparison of experimental transfer characteristics and simulation results for varying channel length. The simulation parameters are shown in Supplementary Table 2.

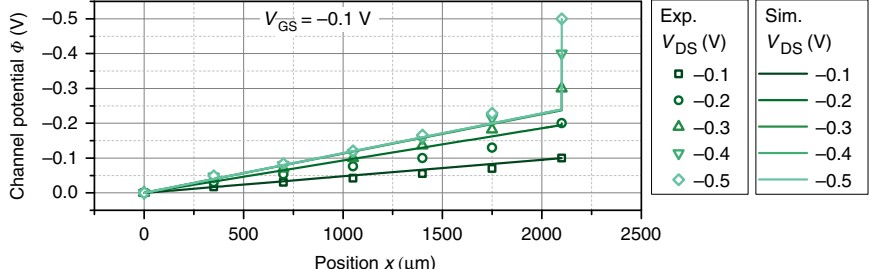

**Fig. 9 Channel potential.** Comparison of the experimentally obtained potential profile along the transistor channel at $V_{GS} = -0.1$ V (symbols) and the calculated potential profile using the parameters summarized in Supplementary Table 2.

agreement with a recent study of Mariani et al.[24], who used scanning electrochemical microscopy to determine the electrochemical potential in the OECT channel. In this microscopic study, a linear dependency of the electrochemical potential inside the channel is found, with abrupt potential changes at the source and drain contacts.

In order to verify the model further, the product of the density of free holes and the charge carrier mobility $p(x)\mu$ as determined by the experiment is plotted in Supplementary Fig. 7 for $V_{GS} = 0.1$ V (open symbols, plots for other gate voltages are provided in Supplementary Fig. 6). The product $p(x)\mu$ is calculated from Ohm's law $j = -ep(x)\mu\frac{d\Phi(x)}{dx}$ and the experimental measurement of $\Phi(x)$.

Assuming that the variation of the hole mobility $\mu$ along the channel is small, the experimentally observed hole concentration can be compared to the normalized hole concentration $\frac{p(x)}{p_0}$ obtained from the 2D model as shown in Supplementary Fig. 7 as well (lines). A good agreement between the hole concentration obtained by the 2D model and the experiment is found. According to Eq. (15), the cation concentration is expected to increase exponentially along the transistor channel, which will lead to a drop in hole concentration close to the drain (Eq. (16), cf. Fig. 7b as well), which is indeed observed in Supplementary Fig. 7.

Despite the good qualitative agreement, the numerical model presented here has to be improved further to become predictive and to be used to extract device parameters reliably. Most importantly, a better description of extraction of holes at the drain contact in the presence of large cation concentrations has to be found. Furthermore, the experimental characterization of OECTs usually shows hysteresis effects[25], which leads to sample to sample variations. Still, the argument used here, that is, that the concentration of ions inside the channel is determined by the equilibrium condition of drift and diffusion currents and not by a gate capacitance, is independent of the particular modeling result.

**Capacitive models as approximation of the equilibrium model.** As discussed above, capacitive OECT models restricting ion currents to the vertical dimension lead to a non-equilibrium state of the device, that is, the ion and potential profile along the channel is not correctly described. Nevertheless, these models were used very successfully in the past and are able to describe the output and transfer characteristic of OECTs.

The good performance of the original model shows that it has to be closely related to the complete 2D model of the device presented here, which takes ion and hole migration inside the channel into account. Indeed, comparing Eq. (16) (derived from equilibrium arguments) and Eq. (1), which is the basis of the original OECT model, one observes that the original device model can be seen as a linear approximation of the full model. In fact, defining a gate capacitance that is exponentially dependent on the

applied potential would lead to identical results. However, the physical interpretation of the two models is different. Whereas in the original model the use of a gate capacitance implies an electrostatic de-doping mechanism, Eq. (16) is derived under the assumption that cations inside the organic semiconductor reach their equilibrium distribution, obtained by enforcing a zero or at least negligible ion current.

Defining a potential dependent gate capacitance is not the only way to correct the original model. Following the results of the 2D model, the potential within the channel (i.e., for all measurements along the channel but excluding the potential applied to the drain electrode) can be fitted by a linear function. An optimized fit is shown in Fig. 10a. The linear fit of the channel potential is extrapolated to the drain electrode, which allows to quantify the additional potential drop caused by the accumulation of ions at the drain $\Delta V_{ion}$. Figure 10b plots this potential drop at the drain contact with respect to the different drain and gate bias. It is observed that the additional potential drop increases for increasing potential difference between the gate electrode and drain electrode, that is, for larger $V_{GS}$ and more negative $V_{DS}$.

The additional potential drop can be formally modeled by a contact resistance, if one divides the potential drop at drain contact by the current flowing through the channel at the same gate and drain bias, that is, $R_C = \frac{\Delta V_{ion}}{I_D}$. Contact resistances were proposed earlier to explain the strong non-monotonic dependency of the transconductance on the gate potential observed in most OECT reports[26].

In Fig. 10c the resulting contact resistance for different drain and gate voltages is plotted. It is found that the contact resistance is exponentially increasing with drain and gate voltages, which is in line with an earlier report[26]. Therefore, the original model as discussed above (cf. Eq. (6)) can be seen as an approximation of the device at low voltages. For larger voltages, the observed redistribution of cations in the channel to reach an equilibrium state can be included by a gate potential dependent contact resistance. This observation explains why the model was used so successfully in the past. The contact resistance as shown in Fig. 10c can as well explain the apparent voltage dependency of the pinch-off voltage $V_P$, that results when the transistors are analyzed using the original model (i.e., using Eqs. (8) or (9)). If the contact resistance $R_C$ becomes significant in comparison to the total resistance of the transistor channel, Eq. (10) has to be corrected for the additional voltage drop across the contact

$$V_P = V_{GS} - V_{DS} - R_C(V_{GS}, V_{DS})I_D = \frac{ep_0}{C_G} - R_CI_D. \quad (18)$$

Therefore, the change of the contact resistance with the gate and drain potential leads directly to a dependency of the pinch-off voltage on the applied potentials.

However, it has to be kept in mind that the addition of a potential dependent contact resistance is only a correction of the

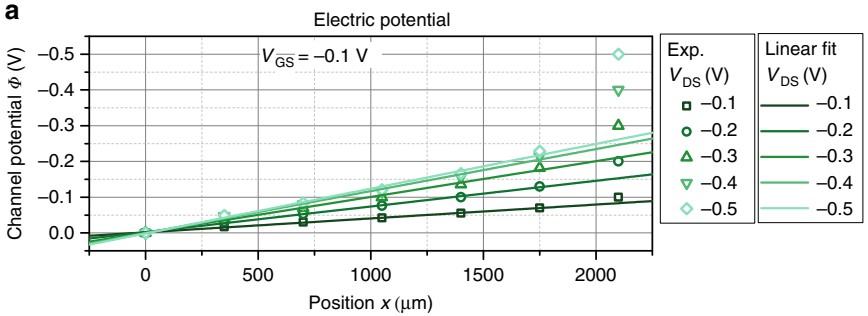

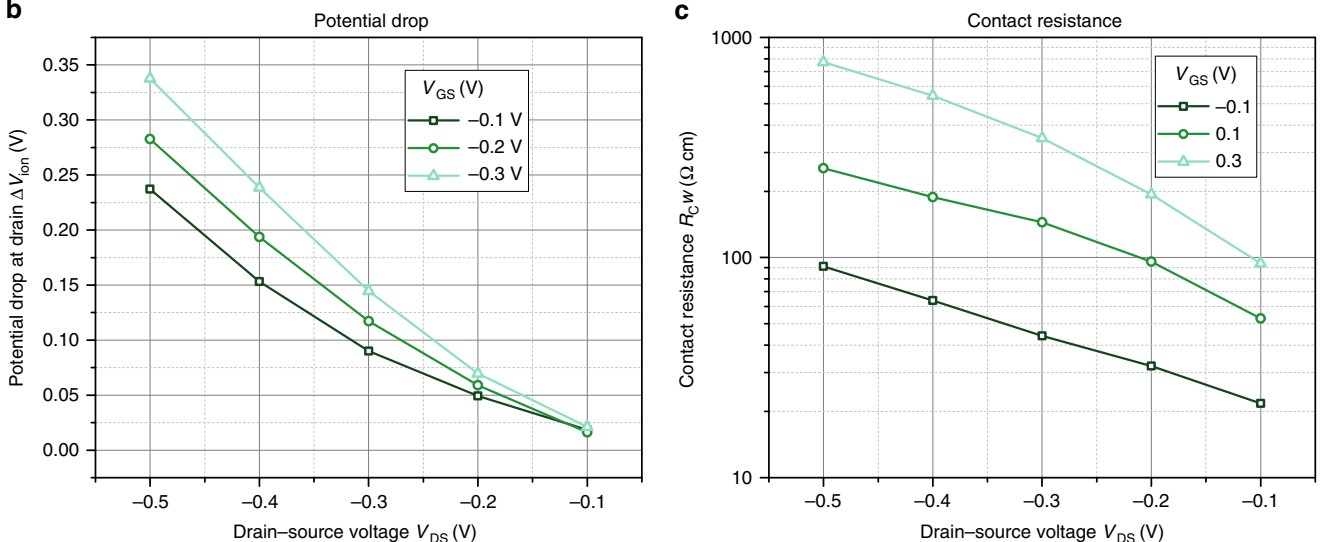

**Fig. 10 Describing ion accumulation by contact resistances. a** Channel potential profile at the gate potential $V_{GS} = -0.1$ V. The lines are linear fits. **b** Potential drop at drain contact for different gate and drain voltages. **c** Contact resistance at varying gate and drain voltages.

conventional model that allows to fit experimental data, keeping all the limitations of the model discussed above, in particular accepting that the ion concentration inside the channel does not represent an equilibrium state. It has furthermore to be stressed that the contact resistance as described here does not represent a contact resistance in the conventional sense, that is, it does not describe an inhibition in the carrier injection. Rather, it describes the effects of ion accumulation at the drain contact, caused by ion currents inside the transistor channel, and the additional potential drop by the accompanying space charge region. Nevertheless, it emphasizes that the drain electrode plays an important role in the working mechanism of OECTs, most likely a more important role than the bulk of the PEDOT:PSS layer.

## Discussion

OECTs have shown very promising results and are discussed as a key technology for the field of organic bioelectronics. Progress in the field was enabled by thin-film transistor device models[3,8,15] that describe ion accumulation and de-doping of the organic semiconductor by a capacitive element included between the gate and the transistor channel.

Despite a qualitative agreement between experimental results and these models, a quantitative analysis of the devices leads to inconsistencies. In particular, the assumption of a capacitive gate coupling leads to an unrealistic distribution of ions inside the transistor channel, and an ion distribution that does not represent a steady-state solution of the device. This, for example, leads to an unrealistic potential distribution along the transistor channel and if used to fit experimental data, results in a potential dependent pinch-off voltage.

An improved numerical device model is presented that consistently solves both, the continuity equation of holes and cations inside the channel of PEDOT:PSS-based transistors. The model shows that the ion concentration increases exponentially toward the drain electrode, leading to a strong drop in hole concentration close to the drain, even before channel pinch off. Furthermore, it predicts that the channel potential inside the transistor channel varies linearly with the position inside the channel and jumps at the drain electrode toward the drain potential.

This altered ion concentration has important consequences for our understanding of operation of the transistor. The model discussed here shifts the focus of the device toward the PEDOT: PSS–drain interface. It shows that a better understanding of injection and extraction of holes at a metal/highly doped organic semiconductor interface in the presence of a high ion concentration is needed to develop an improved device model and to find clear design rules for organic semiconductors used in highly efficient OECTs.

## Methods

**Device fabrication**. The electrodes as shown in Supplementary Fig. 1 are structured by photolithography. To obtain the electrode structure, the photoresist AZ 2020 is spin-coated at 3000 r.p.m. on cleaned glass substrates and subsequently baked at 110 °C on a hot plate. The photoresist is exposed to ultraviolet light by a Karl SUSS Mask Aligner, post baked at 110 °C and developed for 2 min in MIF 300 developer. The metal electrodes are deposited by vacuum deposition of 10 nm chromium, followed by 40 nm of gold onto the photoresist-patterned glass substrates. The source, drain, and gate electrodes are structured by lift-off of the photoresist in acetone. The channel consists of seven contacts with 350 μm gap between each electrode.

Twenty milliliters of PEDOT:PSS (PH1000 provided by Clevios) is mixed with 1 ml ethylene glycol and 50 μl dodecyl benzene sulfonic acid to enhance the

conductivity of the semiconductor. PEDOT:PSS thin films are deposited onto the substrates by spin coating at 1000, 2000, 3000, and 4000 r.p.m. PEDOT:PSS is subsequently baked at 130 °C in a nitrogen-filled glovebox (oxygen and humidity levels below 0.1 p.p.m.) and rinsed with DI water to remove any excess low molecular compounds. The resulting films have an average conductivity $\sigma$ of ~600 S cm$^{-1}$ (cf. Supplementary Fig. 8).

The PEDOT:PSS channel and gate is structured without photolithography. We use a mask to cover the channel and gate area before exposing the PEDOT:PSS layer to oxygen plasma (200W for 2 min, Oxford 80 Plasma Lab) to remove PEDOT:PSS everywhere except in the channel and gate region.

**Preparation of electrolyte**. The preparation of the electrolyte follows the procedure published by Khodagholy et al.[27] C2MIM EtSO$_4$ purchased from Sigma-Aldrich is mixed with 100 mM NaCl in 4:1 ratio to obtain the oom temperature ionic liquids.

**Electrical characterization**. All electrical characterization is carried out by a Keithley 4200 semiconductor analyzer inside a nitrogen-filled glovebox. To ensure that the additional probe contacts within the channel area are not influencing the overall electrical behavior of the OECTs, the output characteristic is measured with and without measuring the channel potential (cf. Supplementary Fig. 9).

## Data availability
Underlying data of all results are provided at https://doi.org/10.21038/blus.2020.0101 and upon request from the authors.

## Code availability
Access to the code is available upon request from the authors.

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

## Acknowledgements
Devices are prepared at the Prototype Facility of the Liquid Crystal Institute, Kent State University. We gratefully acknowledge funding from the National Science Foundation (Grant ECCS 1750011 and ECCS 1709479).

## Author contributions
V.K. and P.R.P. designed, processed, and characterized the OECTs. D.D. and R.K.R.K. characterized the electric performance of devices. B.L. implemented the numeric model, analyzed the calculation results, and supervised the project. All authors have participated in analyzing the results and contributed to writing the manuscript. All authors have given approval to the final version of the manuscript.

## Competing interests
The authors declare no competing interests.
