## [Peer Review File · Nature Communications]

Reviewers' Comments:

Reviewer #1:

Remarks to the Author:

This manuscript by Kaphle et al. presents a numerical device model of organic electrochemical transistors that can better describe the equilibrium state and ion/potential distribution in the channel, showing improvement compared with existing models from thin-film transistor theories. The results highlight the critical role of the channel/drain interface, which may offer useful information for further optimization of OECT devices. However, the authors are suggested to consider the following points:

1. The modeling study here requires more experimental evidence that can consolidate the model independently. For instance, can the ion and potential distributions in the channel be checked by material analysis, e.g. Kelvin Probe Force Microscopy etc., to testify the validity of the model? The significance of the work is affected by the absence of such experimental evidence.
2. The work is essentially based on a 2D drift-diffusion model. However, the presence of ionic and electronic currents may induce thermal effects, which can lead to thermophoresis process as well. This possibility needs to be investigated in this context.
3. The modeling of the organic electrochemical transistors was performed on very large channel lengths, ranging from one to thousands of microns, which may affect the general applicability of the model to nanoscale devices with significantly enhanced electric field strengths. Can authors comment on this?
4. The references of the manuscript seem very poor to me. Relevant and latest progresses in the research field should be included.

Reviewer #2:

Remarks to the Author:

Summary

The paper examines a topic of current interest, a self-consistent 2D description of both charge carriers in OECTs. The numerical study is complemented with a multi-electrode experiment with reasonable agreement. I believe the community would find the results interesting and useful. My only serious concern is that the authors refer to the Bernard's model as the current best understanding of steady-state behavior. The topic has been of much interest since the publication of that paper over 10 years ago and there have been studies published, particularly in the last few years. The authors need to summarize the actual state of the art and make clear how their work advances current understanding.

Minor points

- The authors state that " p_0 is the density of holes in the PEDOT:PSS layer without injected cations, i.e. the doping concentration, which is supposed to equal the density of sulfonate (P S S -) groups." I believe it is typical to assume that there are excess sulfonate groups, only a fraction of which are compensated by holes in the PEDOT. I do not believe this changes the conclusions of the paper.
- Coordinate units for x in figure 6 are missing or hidden by drain.
- Table 1: The symbol ϵ_0 should be just ϵ
- The introduction highlights the observed non-monotonic transconductance with gate voltage as one of the weaknesses of the existing literature. However, I do not see that the body of the paper addresses this topic, so I am unclear what the intent of the introductory material was.

Major point

In the introduction, the authors describe the current state of understanding of the steady-state

conditions of an OECT as the Bernards model which neglects lateral ionic currents within the channel. However, there have been several recent studies that examined this question. For example, three models of ion transport were numerically studied in [1] and compared to a multi-electrode experiment similar to the one reported here. That paper found that global charge neutrality fit the data, paralleling the conclusions of this manuscript. Mobile ions were also studied with a 2D numerical scheme in [2]. Studies have also attempted to experimentally determine relevant quantities such local electrochemical potential (e.g. [3]). Does this paper refute, confirm or extend these existing studies?

References

1. A Shirinskaya et. al, Numerical Modeling of an Organic Electrochemical Transistor, Biosensors 2018
2. M. Z. Szyman'ski et. al, 2-D Drift-Diffusion Simulation of Organic Electrochemical Transistors, IEEE Trans. Electron Devices, 2017
3. F. Mariani et. al, "Microscopic Determination of Carrier Density and Mobility in Working Organic Electrochemical Transistors, Small 2019

Reviewer #3:

None

We would like to thank the reviewers for their time and their thorough assessment of our manuscript. Following their comments, we re-wrote the introductory section to add a detailed review of the current state of OEET device model and to highlight our advancements compared to these models.

In more detail, we would like to respond to the comments raised by the reviewers below. We decided to address the comments of reviewer 2 first, as we will use some of his comments to address comments of Reviewer 1 as well.

Reviewer 2

Comment 1: The paper examines a topic of current interest, a self-consistent 2D description of both charge carriers in OEETs. The numerical study is complemented with a multi-electrode experiment with reasonable agreement. I believe the community would find the results interesting and useful. My only serious concern is that the authors refer to the Bernards model as the current best understanding of steady-state behavior. The topic has been of much interest since the publication of that paper over 10 years ago and there have been studies published, particularly in the last few years. The authors need to summarize the actual state of the art and make clear how their work advances current understanding.

[...]

In the introduction, the authors describe the current state of understanding of the steady-state conditions of an OEET as the Bernards model which neglects lateral ionic currents within the channel. However, there have been several recent studies that examined this question. For example, three models of ion transport were numerically studied in [1] and compared to a multi-electrode experiment similar to the one reported here. That paper found that global charge neutrality fit the data, paralleling the conclusions of this manuscript. Mobile ions were also studied with a 2D numerical scheme in [2].

Reply: We thank the reviewer for his assessment of the manuscript. We agree that our original manuscript has not satisfactorily summarized the status of the field and the progress that has been made since Bernard's model was published. We re-wrote the introduction section to give a more balanced background. In particular, we included ref. [1] and [2] provided by the reviewer.

We would like to stress that the overall conclusion of the manuscript is not altered. Despite the progress made in improving OECT models, all models restrict ion movement to one dimension, which leads to a non-steady state solution of the problem (ref [1] cited by the reviewer does as well restrict ion movement, although more implicit). The only exception is the 2D numerical scheme [2] of Szymanski et al, who as well find discrepancies between their solution and capacitive models (such as the Bernard's model). However, neither is the origin of these discrepancies studied in this manuscript, nor do the authors present a conclusive discussion of the implications of later ion currents in the organic semiconductor.

We tried to further emphasize the novelty of our results in the revised introductory statement as well. In particular, our device model places a much larger focus on the polymer/drain electrode interface compared to any capacitive model. Therefore, further materials design and optimization of OECTs should focus on an improved characterization and modeling of charge extraction at the drain electrode in the presence of high ion densities, which is a major shift compared to current optimization efforts.

The introduction section now reads:

Several models were proposed to quantitatively describe OECT behaviour, in particular the process of doping/de-doping the organic semiconductor [5-12]. The most widely used models separate the device into an ionic and electronic system [13-15]. Ions are assumed to move vertically (i.e. from the gate electrode into the transistor channel, in the following denoted as y-axis), whereas hole transport is restricted to a horizontal movement from source to drain electrode (i.e. along the x-axis). This assumption, resembling the gradual channel approximation of standard thin film theory, allows to calculate the density of ions inside the transistor channel $p_{ion}(x)$ as a function of the difference between the channel potential $\phi(x)$ and the applied gate potential V_{GS} [7, 13, 15]

To be able to derive an analytic description of OECTs, it is often postulated that the density of injected ions $p_{\text{ion}}(x)$ is directly proportional to the potential difference $V_{\text{GS}} - \phi(x)$. Under this assumption, the process of injecting ions into the channel can be described by a capacitive element C_G included between the gate electrode and the PEDOT:PSS channel. In a first version of the model, the capacitance was assumed to scale with device area [7], whereas Rivney et al. found that it depends on the volume of the semiconductor channel [16]. This volumetric gate capacitance reflects the observation that ions are injected into the full volume of the polymer, resulting in huge transconductance values observed frequently [17].

Capacitive models are widely used in the literature [13-15] as they allow to conveniently discuss and analyze transistor results. However, the precise nature of the gate capacitance is still intensively discussed. Some model relax the assumption of a direct proportionality between $V_{\text{GS}} - \phi(x)$ and $p_{\text{ion}}(x)$ and instead calculate the density of injected ions from basic drift-diffusion equations. For example, Shirinskaya et al. used a 1D numerical model to determine $p_{\text{ion}}(x)$ and hence the conductivity of the organic semiconductor as a function of position inside the transistor channel $\sigma(x)$ [18] Coppede et al. proposed an analytical 1D solution for the ionic current injected into PEDOT:PSS under the assumption of a constant electric field inside the electrolyte [19].

However, regardless of the detail with which the density of injected ions $p_{\text{ion}}(x)$ is calculated, ion movement was always limited to one dimension, i.e. to a movement perpendicular to the transistor channel. This assumption, however, has been put into question by recent results of Szymanski, who found that not restricting ion movement inside the transistor to one dimension leads to a different ion concentration $p_{\text{ion}}(x)$ as predicted by capacitive models [10].

Here, we study the applicability of capacitive models by analyzing the electric potential along the transistor channel $\phi(x)$. Our data indicates that these models indeed fail to

describe the steady-state of the transistors. It is shown that the assumption of a gate capacitance leads to ion concentrations inside the transistor channel that would result in significant lateral ion currents. These lateral currents, however, are neglected in capacitive models, forcing the derived solutions into an unrealistic, out-of-equilibrium state. With the help of a 2D drift-diffusion model that solves the continuity equation of holes and cations consistently along the x- and y-direction, it is shown that in contrast to predictions of current OECT models, ions follow an exponential distribution along the transistor channel, which leads to an accumulation of ions at the drain electrode and an additional potential drop at the interface. Overall, the newly found steady state distribution of ions inside the transistor channel shifts the focus to understand details of device operation away from the bulk organic semiconductor to the organic semiconductor/drain electrode interface.

Comment 2: *Studies have also attempted to experimentally determine relevant quantities such local electrochemical potential (e.g. [3]). Does this paper refute, confirm or extend these existing studies?*

We thank the reviewer for drawing our attention to this recent paper. Yes, indeed, the electrochemical potential measured by Mariani et al. reproduces our results well, and we have included the following sentence to our manuscript.

This result is as well in good agreement with a recent study of Mariani et al., who used scanning electrochemical microscopy to determine the electrochemical potential in the OECT channel [24]. In this microscopic study, a linear dependency of the electrochemical potential inside the channel is found, with abrupt potential changes at the source and drain contacts.

Comment 3: *The authors state that “ p_0 is the density of holes in the PEDOT:PSS layer without injected cations, i.e. the doping concentration, which is supposed to equal the density of sulfonate (PSS⁻) groups.” I believe it is typical to assume that there are excess sulfonate groups, only a*

fraction of which are compensated by holes in the PEDOT. I do not believe this changes the conclusions of the paper.

We agree with the reviewer, and changed the sentence to

*where p_0 is the density of holes in the PEDOT:PSS layer without injected cations, i.e. the doping concentration, which is supposed to be **proportional** to the density of sulfonate (PSS) groups.*

This discussion is similar to the discussion of the doping efficiency in small molecular organic semiconductors, and indeed the density of free holes cannot be assumed to equal the density of dopants. However, we would like to stress that this parameter is not changing the overall trends in the simulations.

Comment 4: *Coordinate units for x in figure 6 are missing or hidden by drain.*

We added the correct units to Figure 6.

Comment 5: *Table 1: The symbol ϵ_0 should be just epsilon*

Absolutely, the error was corrected.

Comment 6: *The introduction highlights the observed non-monotonic transconductance with gate voltage as one of the weaknesses of the existing literature. However, I do not see that the body of the paper addresses this topic, so I am unclear what the intent of the introductory material was.*

In this manuscript, we intend to show that a 1D treatment of OECTs leads to a faulty description of OECTs, without focusing on the origin of the peak in transconductance. We changed the introduction section (see above) and removed the section discussing the transconductance and the gate potential dependence of the transconductance. We now only mention this topic later on when discussing contact resistances but tried to put a much weaker focus on this topic.

Nevertheless, we include here some recent simulation results below showing that our model indeed reproduces the gate potential dependency of the transconductance. However, considering the length of the manuscript, we decided not to add these simulations and to discuss the origin of the peak in detail.

Figure 1: Experimental (a) and simulated dependency of the transconductance on the gate potential.

Reviewer 1

Comment 1: *This manuscript by Kaphle et al. presents a numerical device model of organic electrochemical transistors that can better describe the equilibrium state and ion/potential distribution in the channel, showing improvement compared with existing models from thin-film transistor theories. The results highlight the critical role of the channel/drain interface, which may offer useful information for further optimization of OECT devices. However, the authors are suggested to consider the following points:*

Reply: We thank the reviewer for his feedback.

Comment 2: *The modeling study here requires more experimental evidence that can consolidate the model independently. For instance, can the ion and potential distributions in the channel be*

checked by material analysis, e.g. Kelvin Probe Force Microscopy etc., to testify the validity of the model? The significance of the work is affected by the absence of such experimental evidence.

Reply: The submitted manuscript features two independent verifications of the model – in Figure 10, the transfer characteristic of OECTs with channel lengths ranging from 350 ... 2100 μm are compared to modeling results. In Figure 11, the experimentally obtained potential profile along the transistor channel is compared to the model. In particular Figure 11 is not merely a fitting result as the same simulation parameters as deduced from Figure 10 are used.

To address the comment of the reviewer further, we added a comparison between the charge density inside the channel (calculated from the experiment) and the charge density obtained from the simulation as Figure 12 to the manuscript. Again, a good agreement is found between experiment and model without adjusting the original device parameters (summarized in Table 2 of the manuscript).

Figure 2. New Figure 12 of the manuscript.

We added the following discussion to the manuscript:

In order to verify the model further, the product of the density of free holes and the charge carrier mobility $p(x)\mu$ as determined by the experiment is plotted in Figure 12 for $V_{GS} = 0.1$ V (open symbols, plots for other gate voltages are provided in Figure 17 of the supplementary information). The product $p(x)\mu$ is calculated from Ohm's law $j = ep(x)\mu \frac{d\phi(x)}{dx}$ and the experimental measurement of $\phi(x)$.

Assuming that the variation of the hole mobility μ along the channel is small, the experimentally observed hole concentration can be compared to the normalized hole concentration $p(x)/p_0$ obtained from the 2D model as shown in Figure 12 as well (lines). A good agreement between the hole concentration obtained by the 2D model and the experiment is found. According to Equation 15, the cation concentration is expected to increase exponentially along the transistor channel, which will lead to a drop in hole concentration close to the drain (Equation 16, cf. Figure 8b as well), which is indeed observed in Figure 12.

Further validation of the model can be found in the literature. A Scanning Probe Microscopy study of OECTs was recently published by Mariani et al (see reply to comment 2 of reviewer 2). Indeed, the microscopic trend in the electrochemical potential inside the transistor channel is in agreement with our model. We added the following sentence to highlight this fact.

This result is as well in good agreement with a recent study of Mariani et al., who used scanning electrochemical microscopy to determine the electrochemical potential in the OECT channel [24]. In this microscopic study, a linear dependency of the electrochemical potential inside the channel is found, with abrupt potential changes at the source and drain contacts.

In addition to these comments, we would like to stress that our argument does not rest on the numeric model alone but is supported by analytical arguments as well (Figure 5). The 2D model is used to clarify the trends and identify the underlying physical processes, and not, as stated on page 28, to provide a quantitative reproduction of the experiment.

Comment 3: *The work is essentially based on a 2D drift-diffusion model. However, the presence of ionic and electronic currents may induce thermal effects, which can lead to thermophoresis process as well. This possibility needs to be investigated in this context.*

The reviewer is absolutely correct that our model is simplifying and not predictive (yet). We discussed this on page 28, lines 343-350. Thermophoresis effects might be another addition to the model, and we will work to improve on this aspect in the future. Still, we would like to stress that the overall conclusion of the manuscript is not restricted by the absence of thermophoretic effects in the model.

Comment 4: The modeling of the organic electrochemical transistors was performed on very large channel lengths, ranging from one to thousands of microns, which may affect the general applicability of the model to nanoscale devices with significantly enhanced electric field strengths. Can authors comment on this?

The reviewer raises an interesting question. We observed that the convergence of the model improves for smaller devices. Because of the scaling of all parameters in our code, large fields observed in the space charge layers are well resolved and do not lead to numerical instabilities. Overall, we can run our code over several orders of magnitude, from micrometer to millimeter long transistors, i.e. across the generally used channel length range.

Still, we think that we can scale the channel length into the sub micrometer regime as well, in particular considering that the layer thickness is of the range of 100nm only, and that the thickness of the charge accumulation layer at the drain (where most of the potential drops) is in the range of a few nanometer.

We added the following comment to the supplementary information:

All variables (j, p, n, p_{ion}, x, y) are normalized internally, improving the convergence of the code at high charge carrier concentrations and large electric field observed in the space charge layers.

Comment 5: The references of the manuscript seem very poor to me. Relevant and latest progresses in the research field should be included.

Department of Physics
Kent State University

P.O. Box 5190, Kent, OH, 44242, USA E-mail: blussem@kent.edu TEL: +1-330-672-2410 Fax: 330-672-2959

Reply: Yes, we agree. As detailed in the response to reviewer 2, we re-wrote the introductory section to include all relevant literature.

Again, we thank the reviewers for their comments and hope that we have addressed it satisfactorily making the manuscript suitable for publication in Nature Communications.

Reviewers' Comments:

Reviewer #1:

Remarks to the Author:

The authors have performed satisfactory revision and addressed all questions. The manuscript may be accepted for publication.

Reviewer #2:

Remarks to the Author:

I believe the authors have addressed my concerns raised in the initial review.

None